# Analysis of Ethylene Signal Regulating Sucrose Metabolism in Strawberry Fruits Based on RNA-seq

**DOI:** 10.3390/plants13081121

**Published:** 2024-04-17

**Authors:** Jian-Qiang Yu, Zhao-Ting Li, Shen Chen, Hong-Sheng Gao, Li-Xia Sheng

**Affiliations:** College of Horticulture and Landscape Architecture, Yangzhou University, Yangzhou 225009, China; yjqdaniu@163.com (J.-Q.Y.); 18252738353@163.com (Z.-T.L.); chenshen999@126.com (S.C.); hsgao@yzu.edu.cn (H.-S.G.)

**Keywords:** strawberry, ethylene, fruit quality, RNA-seq, sugar

## Abstract

Ethylene is a key hormone that regulates the maturation and quality formation of horticultural crops, but its effects on non-respiratory climacteric fruits such as strawberries are not yet clear. In this study, strawberry fruits were treated with exogenous ethephon (ETH) and 1-methylcyclopropene (1-MCP). It was found that ETH treatment increased the soluble solids and anthocyanin content of the fruits, reduced hardness, and decreased organic acid content, while 1-MCP treatment inhibited these processes. Transcriptome analysis revealed that differentially expressed genes (DEGs) were enriched in the starch–sucrose metabolism pathway. qRT-PCR results further showed significant changes in the expression levels of sucrose metabolism genes, confirming the influence of ethylene signals on soluble sugar accumulation during strawberry fruit development. This study elucidates the quality changes and molecular mechanisms of ethylene signal in the development of strawberry fruits, providing some key targets and theoretical support for guiding strawberry cultivation and variety improvement.

## 1. Introduction

Strawberries (*Fragaria* × *ananassa* Duch.) are perennial herbaceous plants belonging to the *Rosaceae* family and the *Fragaria* genus. The fruits are brightly colored, soft, juicy, and have a sweet and tangy flavor. They are rich in nutrients such as aroma compounds, proteins, organic acids, and essential minerals [1]. Strawberries are considered a model material for studying the ripening mechanisms of non-respiratory climacteric fruits. Fruit ripening is a highly coordinated, irreversible, and genetically programmed process that involves various biochemical, physiological, and sensory modifications. These modifications result in the development of soft, edible, and mature fruits with desirable quality characteristics [2]. This process is accompanied by various quality changes, such as the accumulation of sugars and anthocyanin pigments, a reduction in organic acid content, fruit softening, and an increase in aroma compounds.

The formation of quality in horticultural crops is regulated by multiple hormones, and ethylene has been demonstrated to play a crucial role in climacteric fruits, participating in the transformation of color, flavor, and hardness during fruit ripening [3]. The development and ripening of climacteric fruits such as apples, tomatoes, bananas, and peaches are influenced by ethylene [4] and are determined by their internal gene regulatory networks. The relevant mechanisms have been thoroughly elucidated [5,6]. Although there is a lack of ethylene release peak during the ripening process of non-climacteric fruits, ethylene still plays an important role in their quality formation [7]. As a typical non-climacteric fruit, strawberries have relatively low levels of endogenous ethylene [8], but they still exhibit a certain pattern during fruit ripening, with higher levels during the green fruit stage, decreased levels during the white fruit stage, and a subsequent increase during the mature red fruit stage [9]. Interestingly, this final increase is similar to the enhanced respiration rate observed in climacteric fruits at the onset of ripening [10]. Transcriptomic and metabolomic studies of transgenic strawberries with reduced ethylene sensitivity, specifically silencing *FaEXP2*, have shown that ethylene is a necessary endogenous hormone for normal strawberry fruit development [11]. However, the specific role of ethylene in strawberry fruit development still requires further investigation.

It is well known that sugar content is an important quality trait in horticultural crops, and the sugar content in fleshy fruits is typically high. Soluble sugars in strawberry fruits mainly include sucrose, glucose, and fructose, with a total content reaching nearly 500 mg/g DW [12]. The sugar content in strawberries is mainly influenced by inter-tissue transport and internal fruit metabolism processes, regulated by hormones and environmental signals [13]. The injection of exogenous ABA into the fruits can increase sucrose content by upregulating the expression of sucrose transporter gene 1 (*FaSUT1*) [14]. However, the role of ethylene in regulating sugar accumulation in strawberry fruits is still unclear.

In this study, we conducted exogenous ethylene (ETH) and ethylene inhibitor (1-MCP) treatments on ‘*Benihoppe*’ fruits at the white fruit stage. The results revealed that ethylene signaling accelerated the ripening process of strawberry fruits, leading to significant changes in the expression levels of numerous genes involved in sucrose and starch metabolism pathways, as well as alterations in sucrose metabolism. Ultimately, it was demonstrated that ethylene signaling is a positive regulatory factor for sugar accumulation during strawberry fruit ripening. This finding provides theoretical support for understanding the impact of ethylene signaling on the ripening and quality formation of non-climacteric fruits and offers crucial targets and guidance for molecular breeding efforts in non-climacteric fruits such as strawberries.

## 2. Results

### 2.1. Ethylene Signal Can Affect the Ripening and Quality Formation of ‘Benihoppe’ Fruit

To investigate whether ethylene plays a critical role in the ripening and quality formation of strawberry fruits, we selected ‘Benihoppe’ fruits at the white fruit stage attached to strawberry plants as experimental materials. The fruits were immersed in different concentrations of exogenous ethylene (ETH) and ethylene inhibitor (1-MCP), while a water immersion treatment was used as a control.

The results revealed that ETH treatment significantly accelerated the ripening of strawberry fruits, with the most pronounced effect observed at a concentration of 2 mM of ethylene (Figure 1A). Conversely, 1-MCP treatment markedly inhibited fruit ripening (Figure 1A). Further analysis of soluble solids, anthocyanin content, hardness, and organic acid content in the fruits showed that ETH treatment led to a significant increase in anthocyanin content (Figure 1C). The soluble solids also increased but not significantly (Figure 1B), while acid content significantly decreased compared to the control group (Figure 1D). The hardness also decreased but not significantly (Figure 1E). In contrast, 1-MCP treatment showed opposite effects (Figure 1B–E). These findings indicate that ethylene signaling is involved in the ripening process of strawberry fruits and regulates the formation of related quality traits.

### 2.2. Transcriptome Analysis of Ethylene Treatment

#### 2.2.1. Sequencing Statistics and Quality Control

To elucidate the underlying mechanisms of ethylene signaling in the ripening process of strawberry fruits, we selected strawberry fruits treated with the most significant effect of 2 mM ethephon, strawberry fruits treated with 1-methylcyclopropene (1-MCP), and control samples for RNA-seq analysis. A total of nine samples were subjected to eukaryotic transcriptome (RNA-seq) analysis. Differential gene expression was determined using the criteria of fold change ≥ 2 and *p* value ≤ 0.05. A total of 55.19 Gb of clean data were obtained, with each sample yielding approximately 5.75 Gb of clean data. The Q30 base percentage was 93.69% or higher for all samples (Table 1).

Each sample’s clean reads were aligned to the specified reference genome, with alignment efficiencies ranging from 91.27% to 92.84%. A total of 108,087 genes were detected. Using the StringTie software (StringTie 1.2.1-v-B), the mapped reads were assembled based on the selected reference genome sequence. The assembled transcripts were then compared with the existing genome annotation information to identify previously unannotated transcription regions, resulting in the discovery of 1931 new genes, with functional annotations obtained for 973 of them. In the CK vs. ETH group, 2346 differentially expressed genes were identified, including 1099 upregulated genes and 1247 downregulated genes (Figure 2). In the CK vs. 1-MCP group, 833 differentially expressed genes were identified, with 240 genes upregulated and 593 genes downregulated (Figure 2).

#### 2.2.2. GO and KEGG Pathway Analysis

To elucidate the specific functions of ethylene signaling in regulating strawberry fruit quality formation, gene ontology (GO) enrichment analysis was performed using the GO database. In the CK vs. ETH group (Figure 3A), through the agriGO website, it was found that the differentially expressed genes (DEGs) were enriched in 25 GO pathways. The top 20 GO pathways were selected based on FDR (≤0.05) for bubble chart visualization. It was observed that a large number of genes were enriched in the “metabolic process” (GO:0008152), “cellular process” (GO:0009987), and “cellular biosynthetic process” (GO:0044249) pathways, with 736, 625, and 260 genes enriched, respectively. In the CK vs. 1-MCP group (Figure 3B), through the agriGO website, it was found that the DEGs were enriched in 190 GO pathways. The top 20 GO pathways were selected based on FDR for bubble chart visualization. It was observed that a large number of genes were enriched in the “single-organism process” (GO:0044699), “nucleotide binding” (GO:0000166), and “single-organism metabolic process” (GO:0044710) pathways, with 143, 104, and 94 genes enriched, respectively.

Additionally, we conducted KEGG enrichment analysis using bioinformatics databases and constructed KEGG pathway enrichment maps. In the CK vs. ETH group, 2346 differentially expressed genes were enriched in 44 KEGG metabolic pathways. The top 20 metabolic pathways were selected based on the *p* value for the construction of the KEGG enrichment bubble chart. From Figure 3C, it can be observed that the DEGs were mainly concentrated in the phenylpropanoid biosynthesis pathway (KO:00940), starch and sucrose metabolism pathway (KO:00500), and flavonoid biosynthesis pathway (KO:00941). In the CK vs. 1-MCP group, 833 differentially expressed genes were enriched in 137 KEGG metabolic pathways. The top 20 metabolic pathways were selected based on the *p* value for the construction of the KEGG enrichment bubble chart. From Figure 3D, it can be observed that the DEGs were mainly concentrated in the pyruvate metabolism pathway (KO:00620), starch and sucrose metabolism pathway (KO:00500), and glycolysis/gluconeogenesis pathway (KO:00010).

### 2.3. Differential Gene Analysis of Starch and Sucrose Metabolic Pathway

By examining the KEGG metabolic pathways in the CK vs. ETH and CK vs. 1-MCP groups, we found that a large number of differentially expressed genes in response to ETH and 1-MCP treatments were enriched in the starch and sucrose metabolism pathway. Therefore, we focused on the differentially expressed genes in the starch and sucrose metabolism pathway and found that several metabolic pathways were influenced by ethylene signaling (Figure 4, Table 2). β-fructofuranosidase (INV) catalyzes the breakdown of sucrose into fructose and glucose within fruit cells. Three β-fructofuranosidase genes (gene 30–32) showed significantly decreased expression levels under ethylene and 1-MCP treatments. UDP-glucose, which catalyzes sucrose regeneration, is converted to fructose-6-phosphate, trehalose-6-phosphate, and glucose by sucrose phosphate synthase (SPS), trehalose-6-phosphate synthase (TPS), and endoglucanase (EGLC), respectively. The SPS genes (gene 11–13), TPS genes (gene 22–25), and EGLC gene (gene 29) involved in these processes showed significantly decreased expression levels after ethylene treatment. UDP-glucose is converted to cyclodextrin by endoglucanase and further converted to glucose by β-glucosidase. The endoglucanase gene (gene 1) and β-glucosidase genes (gene 16–18) involved in these processes showed lower gene expression levels in both the ethylene and 1-MCP treatments compared to the CK group. The β-glucosidase genes (gene 15, gene 19) showed lower expression levels than the CK group under ethylene treatment, while their expression levels were higher than the CK group under 1-MCP treatment. Trehalose-6-phosphate is converted to trehalose and eventually to glucose by trehalose-6-phosphate phosphatase (TPP). The TPP genes (gene 3, gene 4) involved in this process showed lower expression levels under ethylene treatment compared to the CK group and slightly similar expression levels under 1-MCP treatment. UDP-glucose is converted to starch by granule-bound starch synthase (WAXY) and a series of enzymes, and maltose is produced by the action of β-amylase, resulting in the generation of glucose. The β-amylase genes (gene 9, gene 10) involved in this process showed lower expression levels under both ethylene and 1-MCP treatments compared to the CK group. UDP-glucose is converted to glucose-6-phosphate via α-D-glucose-1-phosphate by phosphoglucomutase (pgm), resulting in the generation of glucose. The phosphoglucomutase genes (gene 26–27) involved in this process showed significantly higher expression levels under ethylene treatment, while the expression levels under 1-MCP treatment were slightly higher than the CK group. Gene 28 showed slightly higher expression levels under ethylene treatment compared to the CK group, and its expression level under 1-MCP treatment was similar to the CK group.

From Figure 5A, it can be seen that there are four genes shared between the CK vs. ETH and CK vs. 1-MCP treatment groups in the starch and sucrose metabolism pathway: gene 9 (maker-Fvb5-1-augustus-gene-130.60), gene 10 (maker-Fvb7-4-augustus-gene-113.46), gene 17 (maker-Fvb3-2-augustus-gene-265.38), and gene 27 (maker-Fvb7-1-augustus-gene-282.41). Gene 9 and gene 10 belong to the β-amylase gene, gene 17 is the β-glucosidase gene, and gene 27 belongs to the phosphoglucomutase gene. They are key enzymes in the sucrose metabolism pathway, regulating the conversion of sucrose to glucose and starch. From the fluorescence quantification results in Figure 5B–E, it can be observed that the expression levels of gene 9, gene 10, and gene 17 decreased under both ethylene and 1-MCP treatments compared to CK. On the other hand, the expression level of gene 27 increased under both ethylene and 1-MCP treatments compared to CK. The results of qRT-PCR analysis confirmed the reliability of the transcriptome sequencing results.

### 2.4. Effect of Ethylene Treatment on Soluble Sugar in Strawberry Fruit

To further clarify the changes in soluble sugars in strawberry fruits after ethylene treatment, the main three sugars, sucrose, glucose, and fructose, were identified and quantified using HPLC in each treatment group. In Figure 6, it can be observed that the soluble sugar content (the sum of sucrose, fructose, and glucose) in strawberries treated with 0.2 mM ethephon was roughly comparable to the control group (CK). The soluble sugar content in strawberries treated with 2 mM ethephon was significantly higher than the CK group, while the strawberries treated with 10 mM ethephon and 1-MCP showed significantly lower soluble sugar content than the CK group (Figure 6). It can be found that the content of sucrose, fructose, and glucose in strawberries treated with 0.2 mM ethephon was not significantly different from the CK group (Figure 6). The content of fructose and glucose in strawberries treated with 2 mM ethephon was significantly higher than the CK group, and the content of sucrose treated with 2 mM ethephon was non-significant.

In Figure 6, it can be found that the glucose content in strawberries treated with 10 mM ethephon was significantly lower than the CK group. But the content of fructose and sucrose in strawberries treated with 10 mM ethephon was not significantly different from the CK group. The content of glucose and sucrose in strawberries treated with 1-MCP was significantly lower than the CK group. The fructose content in strawberries treated with 1-MCP was also lower than the CK group but non-significant. These data further illustrate that an appropriate concentration of ethephon (2 mM) can increase the soluble sugar content in strawberries, while 1-MCP treatment decreases the soluble sugar content.

## 3. Materials and Methods

### 3.1. Transcriptome Analysis of Ethylene Treatment

We selected evenly shaped and sized white-stage ‘*Benihoppe*’ strawberry fruits (*Fragaria* × *ananassa* Duch.) from Xijiang Ecological Park in Guangling District, Yangzhou City, Jiangsu Province, China (longitude: 119.40, latitude: 32.40). The Xijiang Ecological Park adopts a glass greenhouse cultivation method with a humidity of 65% and day–night temperatures of 25 °C/15 °C. The strawberry fruits were immersed in 0.2 mM, 2 mM, and 10 mM concentrations of ethephon (ETH), 150 ppm of 1-methylcyclopropene (1-MCP, as an ethylene inhibitor), and water (as a control) for 1 min each day, for a total duration of 7 days.

The measurements were repeated three times for each indicator, and three strawberries were reused each time.

### 3.2. Physiological Index Determination

#### 3.2.1. Anthocyanin Content Determination

The PH differential method was used. Weigh 1.0 g of strawberry puree and use acid ethanol [V (99% anhydrous ethanol)/V (0.2 mol/L hydrochloric acid) = 3:2] as the extraction solution with a sample-to-solvent ratio of 1:10 (g/mL). After dissolving the sample, extract it in a 50 °C water bath for 60 min, and centrifuge the extraction solution for 20 min at a speed of 4000 rpm. Take the supernatant and dilute it 10 times with potassium chloride buffer solution at pH 1.0 and sodium acetate buffer solution at pH 4.5, respectively. Allow it to equilibrate for 110 min and measure the absorbance of the diluted solution at wavelengths of 525 nm and 700 nm. Calculate the anthocyanin content according to the following formula.
$$\mathrm{Anthocyanin}\left(\mathrm{mg}/L\right)=\frac{A*\mathrm{Mw}*\mathrm{DF}*1000}{\varepsilon *L}$$
A=(A525 − A700)pH = 1 − (A525 − A700)pH = 4.5

In the equation, MW represents the molecular weight of delphinidin-3-glucoside, which is 449.2 g/mol; DF represents the dilution factor; ε represents the molar absorptivity of delphinidin-3-glucoside, which is 26,900 L/mol·cm; and L represents the path length.

#### 3.2.2. Determination of Fruit Hardness and Soluble Solids

The hardness of the strawberry fruit was measured in the middle section using an Edibor HP-50 digital push–pull force gauge (Shenzhen, China), and the data were recorded. The measurements were repeated three times, with three strawberry fruits used for each repetition.

The soluble solids were determined using an ATAGO Pocket PAL-1 handheld refractometer (Shanghai, China). The strawberry fruit tip was squeezed to extract the juice, which was then placed on the handheld refractometer to measure the soluble solids. The measurements were repeated three times, with three strawberry fruits used for each repetition.

#### 3.2.3. Determination of Total Acid Content

Weigh 0.5 g of fresh strawberry sample and grind it thoroughly. Add a small amount of distilled water free of carbon dioxide. Heat the mixture in a 75 °C water bath for 0.5 h. After cooling, make up the volume to 20 mL. Filter the solution using dry filter paper and collect the filtrate for further use. Accurately pipette 5 mL of the filtrate, add 2–3 drops of phenolphthalein indicator, and titrate with 0.1 mol/L standard KOH solution until a faint pink color persists for 30 s without fading. Record the volume used. Calculate the total acid content in the sample using the following formula.
$$X=\frac{C*X\left(V_{1}-V_{0}\right)*70}{V_{2}}$$

In the equation: X represents the mass concentration of total acid in the sample (expressed as citric acid) in g/L; C represents the concentration of the standard sodium hydroxide solution in mol/L; V_1_ represents the volume of NaOH standard solution consumed in mL; V_0_ represents the volume of blank control NaOH standard solution consumed in mL; V_2_ represents the volume of the sample taken in mL; and 70 represents the molar mass of citric acid in g/mol.

### 3.3. RNA-seq Analysis

#### 3.3.1. Transcriptome Library Construction

We selected strawberry fruits treated with 2 mM ETH, 1-MCP, and water as CK for RNA-seq analysis. The library preparation reagent kit used for library construction is the NEBNext Ultra RNA Library Prep Kit for Illumina (NEB, Ipswich, MA, USA), following the instructions provided with the kit. After quality control of the libraries, pooling is performed based on the effective concentration of each library and the desired amount of data. Finally, sequencing is performed using the Illumina HiSeq 2100 platform (Shenzhen, China).

#### 3.3.2. Transcriptome Data Analysis

Data Quality Control: Raw data (raw reads) obtained from the sequencing run are processed by removing reads containing adapters and low-quality reads (reads with an N content greater than 10% or with more than 50% of bases having a quality value less than 10), resulting in clean reads. Gene expression quantification analysis: A genome-wide index of the octoploid strawberry genome is constructed using HISAT2 (Nanjing, China). Each library’s clean reads are aligned to the genome, and the read count for each gene (read counts) is calculated. The read counts are then normalized to obtain expression levels using the FPKM (Fragments Per Kilobase of transcript per Million fragments mapped) method, which serves as a measure of gene expression. The FPKM calculation formula is as follows:$$\mathrm{FPKM}=\frac{\mathrm{cDNA}\;\mathrm{Fragments}}{\mathrm{MappedFragments}\left(\mathrm{Millions}\right)\cdot \mathrm{TranscriptLength}\left(\mathrm{kb}\right)}$$

In the equation, cDNA Fragments represents the number of fragments aligned to a specific transcript, i.e., the number of paired-end reads. Mapped Fragments (Millions) represents the total number of fragments aligned to the transcript, measured in millions (10^6^). Transcript Length (kb) refers to the length of the transcript, measured in kilobases (10^3^ bases). Differential expression analysis: DESeq2 is used for differential expression analysis, with a filtering threshold of fold change ≥ 2 and *p* value ≤ 0.05 as the criteria for selection.

### 3.4. Determination of Soluble Sugar (Sucrose, Fructose, and Glucose) Content

Preparation of soluble sugar solution: take the fruits of each treatment and maturity stage for liquid nitrogen research, weigh 3 g in a 10 mL centrifuge tube, cool to room temperature, centrifuge at 12,000 rpm for 10 min, take the supernatant, and set the volume to 10 mL. A total of 1 mL of the solution was filtered using a 0.22 μm aqueous filter membrane, and the filtrate was collected for testing.

HPLC was used to detect the soluble sugar (sucrose, fructose, and glucose) content of strawberry fruit. Chromatographic conditions: the temperature of the detection cell was 75 °C, the column temperature of the column was 35 °C, the model was 150 mm × 46 mm × 5 μm, the mobile phase was deionized water and methanol, the flow rate was 1 mL/min, each injection was 10 μL, and three replicates were performed.

The formulation of standard working curves for soluble sugar: 10 mg of sucrose, fructose, and glucose were dissolved in 5 mL of deionized water to obtain 2000 ppm of monosaccharide mother liquor, and 6 working solutions with different concentrations (1200 ppm, 1000 ppm, 800 ppm, 600 ppm, 500 ppm, 250 ppm) were configured. The regression equation was obtained by using the standard concentration as the independent variable and the peak area as the dependent variable (Table 3).

### 3.5. qRT-PCR Analysis

The Kingsray Biotechnology Real-time PCR Primer Design online tool was used for designing fluorescence quantitative PCR primers. ChamQ SYBR Color qPCR Master Mix (Nanjing, China) and CFX96 Real-time PCR instrument (Nanjing, China) were used for quantitative analysis of gene expression for gene 9, gene 10, gene 17, and gene 27. FaActin was used as the internal reference gene for the normalization of expression levels (Table 4).

### 3.6. Statistical Analysis

All experiments were performed in triplicate. Error bars show the standard deviation of three biological replicates. Significant differences were detected by *t*-test using GRAPHPAD PRISM 6.02 software (ns, non-significant; *, *p* < 0.05; **, *p* < 0.01).

## 4. Discussion

The development and ripening of horticultural crop fruits are influenced by external environmental conditions such as temperature and light, as well as being regulated by endogenous plant hormones. Among them, ethylene, as an important plant hormone, plays a crucial role [15]. In climacteric fruits, ethylene is considered the main hormone involved in regulating fruit ripening, and it is regulated by other plant hormones or signaling molecules, such as ABA (abscisic acid) signaling [16], auxin signaling [17,18], JA (jasmonic acid) signaling [19], and NO (nitric oxide) signaling [20], among others.

Compared to climacteric fruits, non-climacteric fruits such as strawberries, grapes, citrus fruits, and melons do not exhibit a sharp increase in respiration during the ripening process, and the release of ethylene is maintained at a relatively low level. Grape fruits show a peak in ethylene release at the onset of ripening, and after treatment with 1-MCP (1-methylcyclopropene, an ethylene signaling inhibitor), there is a decrease in sucrose content, delayed fruit enlargement, and anthocyanin accumulation [21]. Application of exogenous ethylene to strawberry fruits affects post-harvest quality and storage life, reducing fruit firmness and accelerating color change. Conversely, exogenous treatment with 1-MCP affects the normal ripening of strawberry fruits [22]. These results indicate that ethylene indeed plays an important role in the ripening process of non-climacteric fruits [23]. Our study also found that for typical non-climacteric fruit, exogenous ethephon treatment significantly promoted fruit ripening at the white-fruit stage, while 1-MCP treatment inhibited this process. In conclusion, ethylene signaling plays a similar role in the ripening process of climacteric and non-climacteric fruits, accelerating the formation of fruit quality. However, due to the significant difference in ethylene release level between the two fruit types, its specific role may need to be further explored.

Abscisic acid (ABA) has been recognized to play an important role in the ripening of non-climacteric fruits [24,25,26]. It can promote softening and pigment deposition in strawberry fruits [27,28]. Injecting exogenous ABA into strawberries can increase the expression of sucrose transporter protein 1 (*FaSUT1*) and increase sucrose content, thereby affecting sugar accumulation in the fruits [14]. On the other hand, auxin acts as an inhibitor of ripening in non-climacteric fruits [29]. The concentration of auxin (IAA) reaches its maximum during the strawberry flower and young fruit stages and then decreases [30]. Gibberellins (GA) have also been observed to inhibit strawberry fruit ripening [31]. Cytokinins are involved in strawberry flower differentiation, increasing the number of flowers and fruit formation [32]. The concentration of the active cytokinin zeatin sharply increases before flowering and then decreases [33]. It has been reported that there is an interaction between ethylene and ABA. In post-harvest strawberry fruits, ethylene treatment can promote ABA accumulation, malic acid metabolism, and a reduction in fruit weight [34]. This indicates that there may be crosstalk between ethylene and ABA signaling, and both play a cooperative role in regulating the quality formation during strawberry fruit ripening. In summary, plant hormones such as ABA, auxin, cytokinins, and gibberellins have certain effects on strawberry ripening and quality formation. However, the specific mechanisms of ethylene’s role in strawberry development are not yet clear. This study found that the application of exogenous ethylene to strawberries resulted in an increase in sugar and anthocyanin content, a decrease in acidity and firmness, and accelerated fruit ripening. However, knowledge of whether the external application of ethephon can affect the content changes in other hormones, and how ethylene signals in tandem with other hormone signals synergically regulate the ripening mechanism of strawberry fruit, are still lacking, which is the focus of further research.

In the future, a comprehensive study will be carried out for different degrees of ethylene signaling. In RNA-seq, it was observed that a large number of genes were enriched in the metabolic process by GO enrichment analysis. This indicates that ethylene signaling plays an important role in regulating the metabolic process of strawberry fruit. Further KEGG analysis showed that DEGs were enriched in starch and sucrose metabolic pathways in both CK vs. ETH and CK vs. 1-MCP groups. By observing the color sucrose and starch metabolic pathway map, we can find the location of differential genes, which promote the conversion of sucrose to fructose and glucose, which is consistent with the determination of soluble sugar content. This study further confirms the key role of the plant hormone ethylene in the ripening and quality formation of non-climacteric fruits such as strawberries. Different concentrations of ethephon have different effects on the ripening process of strawberries. Due to the complexity of the action mechanism of the ethylene signal, only 0.2 mM of the optimum concentration of ethephon was selected in this study, which has certain limitations.

Strawberries are commonly used as a model crop for studying non-climacteric fruits. The initiation of fruit ripening in strawberries requires only a small amount of ethylene, and the production of ethylene exhibits characteristic patterns at different stages of strawberry fruit development. Specifically, there is a fluctuation but overall low level of ethylene release during the small green fruit to large green fruit stage. The ethylene content sharply increases from the white fruit stage to the color transition stage, then rapidly decreases and stabilizes at a lower level. Finally, it increases again during the red ripening stage [10]. Despite the very low concentration of endogenous ethylene in strawberry fruits, ethylene plays an important role in the ripening process of strawberry fruits [21,29]. In this study, by treating strawberry fruits at the white fruit stage with different concentrations of exogenous ethylene, it was found that 2 mM ethylene promoted the accumulation of soluble sugars such as sucrose, while low concentration ethephon (0.2 mM) had little effect, and high concentration ethephon (10 mM) and 1-MCP affected the accumulation of total soluble sugars. This study demonstrates that ethylene signaling indeed participates in the strawberry fruit ripening process and directly regulates the accumulation of sugars, thus enriching the key role of ethylene in strawberry cultivation and providing theoretical guidance for relevant molecular breeding work.

## 5. Conclusions

The role of ethylene in the development of non-respiratory burst-type fruits is not yet clear. In this experiment, strawberry fruits were treated with various concentrations of ethephon and ethylene inhibitor (1-MCP), and significant changes in fruit quality were observed. Following ethephon treatment, the soluble solid content and anthocyanin content increased compared to the control group (CK), while the firmness and organic acid content decreased. Among the treatments, the effect was most significant with 2 mM ethephon. On the other hand, 1-MCP treatment significantly inhibited fruit ripening and quality formation. Transcriptome sequencing analysis revealed a large number of differentially expressed genes enriched in the starch and sucrose metabolism pathway, and this result was further confirmed using qRT-PCR.

In conclusion, we found that the soluble sugar content of strawberry fruits underwent significant changes after treatment. Specifically, 2 mM ethephon promoted the accumulation of soluble sugars, including sucrose, while the content of soluble sugars decreased after 1-MCP treatment, which inhibits the ethylene signal. Overall, our study demonstrates the important role of ethylene signaling in regulating strawberry fruit ripening and sugar accumulation. We also identified four key genes, providing gene targets and guidance for further exploration of the role of ethylene signaling in non-respiratory burst-type fruits.

## Figures and Tables

**Figure 1 plants-13-01121-f001:**
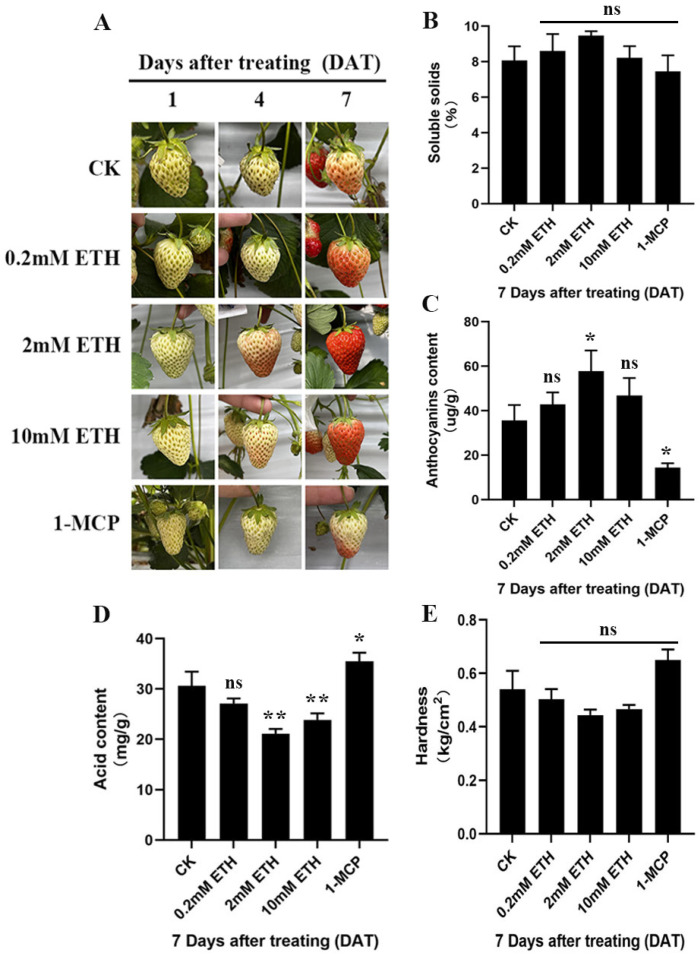
Ethylene signal accelerated the ‘Benihoppe’ fruit quality formation and ripening process. After treatment with water (CK), different concentrations of ethephon (0.2 mM, 2 mM, and 10 mM), and 1-MCP (ethylene signaling inhibitor), (**A**) fruit phenotype chart, (**B**) fruit soluble solids content, (**C**) fruit anthocyanin content, (**D**) fruit organic acid content, and (**E**) fruit firmness were obtained. ns, non-significant; significance * and ** at 0.01 < *p* ≤ 0.05 and *p* ≤ 0.01, respectively.

**Figure 2 plants-13-01121-f002:**
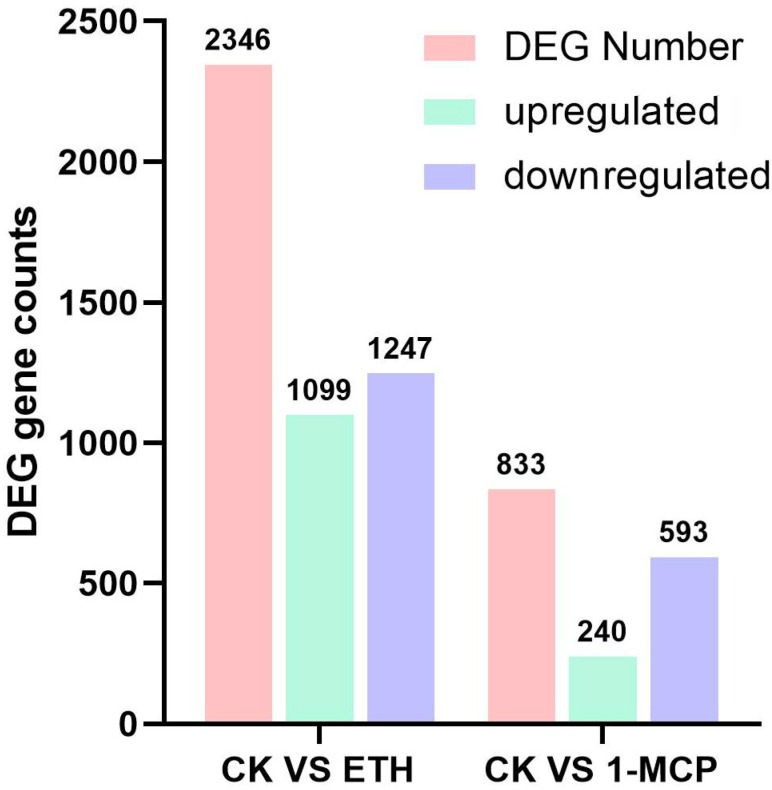
The number of DEGs (differentially expressed genes) between the CK vs. ETH treatment groups, and the CK vs. 1-MCP treatment groups.

**Figure 3 plants-13-01121-f003:**
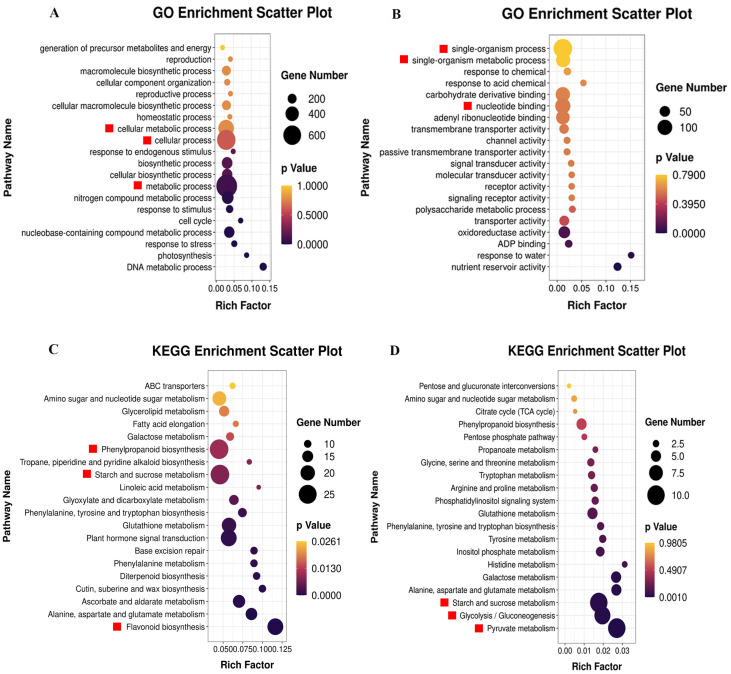
The GO and KEGG enrichment analysis of DEGs (differentially expressed genes). (**A**) CK vs. ETH DEGs GO enrichment analysis. (**B**) CK vs. 1-MCP GO enrichment analysis. (**C**) CK vs. ETH DEGs KEGG enrichment analysis. (**D**) CK vs. 1-MCP KEGG enrichment analysis. The red boxes are the top 3 pathways in the total number of DEGs.

**Figure 4 plants-13-01121-f004:**
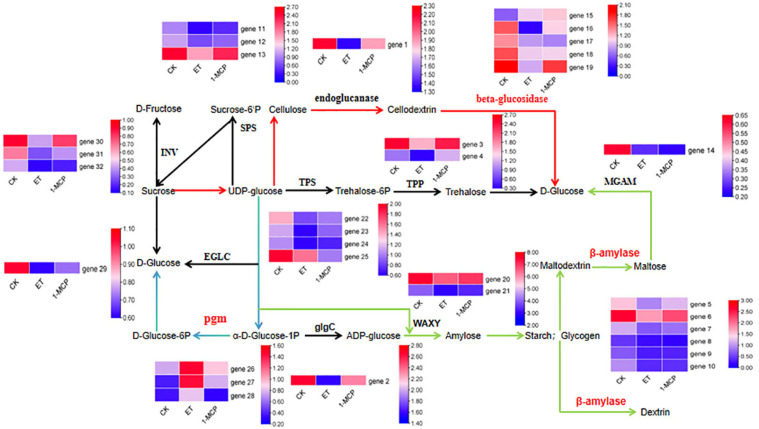
Color pathway map of DEGs (differentially expressed genes) in starch–sucrose metabolic pathway. The difference-expressed genes shared by CK vs. ETH and CK vs. 1-MCP groups were marked with different color lines: red lines: β-glucosidase plays a role in promoting sucrose to produce glucose; green lines: beta-amylase plays a role in the production of starch and glucose from sucrose; blue lines: pgm works to produce glucose from sucrose; black lines: others.

**Figure 5 plants-13-01121-f005:**
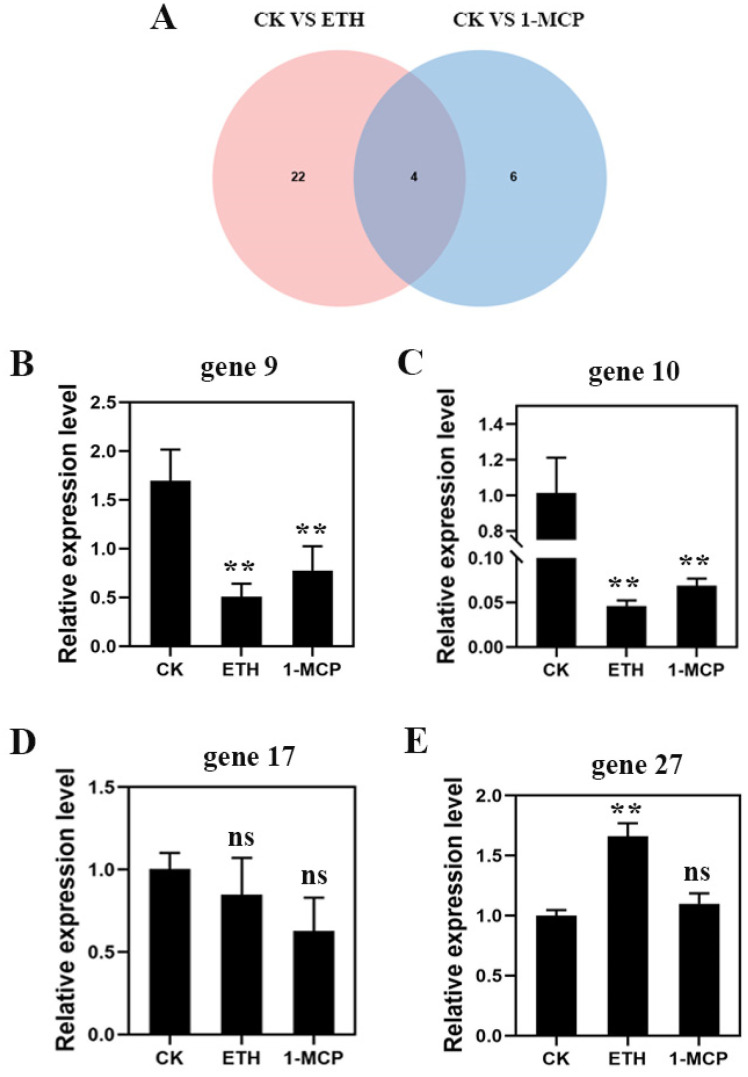
Venn diagram of common differential genes between the CK vs. ETH and CK vs. 1-MCP treatment groups (**A**) and differential gene expression of (**B**) gene 9, (**C**) gene 10, (**D**) gene 17, and (**E**) gene 27. ns, non-significant; significance ** at *p* ≤ 0.01, respectively.

**Figure 6 plants-13-01121-f006:**
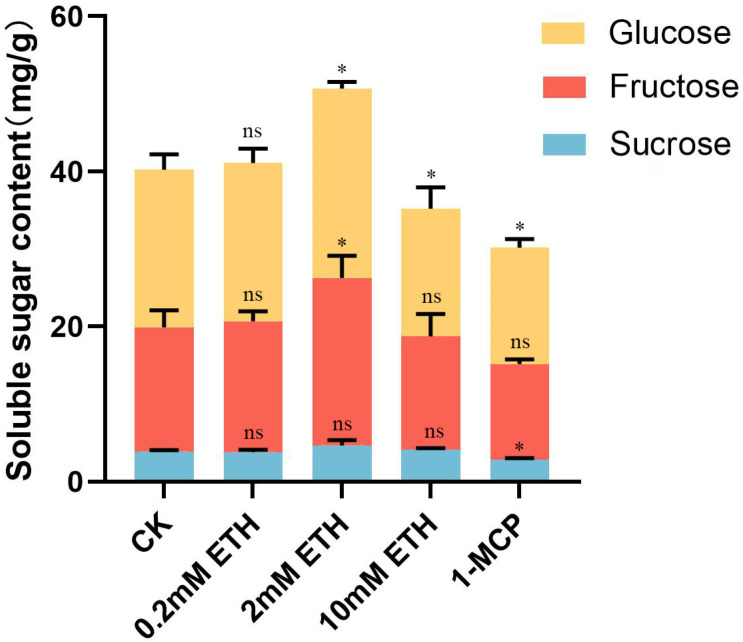
Ethylene signaling promoted the accumulation of soluble sugars in strawberry fruit. The soluble sugar content (the sum of sucrose, fructose, and glucose) of strawberries after water (CK), different concentrations of ethephon (0.2 mM, 2 mM, and 10 mM), and 1-MCP (ethylene signaling inhibitor) processing. ns, non-significant; significance * at 0.01 < *p* ≤ 0.05.

**Table 1 plants-13-01121-t001:** Quality assessment of individual samples in the transcriptome.

Sample ID	Read Sum	Base Sum	GC (%)	N (%)	Q20 (%)	Q30 (%)
CK-1	19,184,077	5,745,561,076	46.54	0	98.21	94.8
CK-2	19,740,638	5,912,302,492	46.43	0	98.16	94.65
CK-3	20,126,908	6,027,476,780	46.59	0	97.73	93.69
M-1	21,968,697	6,578,189,556	46.46	0	98.43	95.42
M-2	20,270,368	6,069,493,864	46.7	0	98.06	94.49
M-3	20,371,609	6,101,097,966	46.42	0	98.08	94.42
Y-1	21,581,232	6,464,065,172	46.27	0	98.11	94.37
Y-2	20,591,704	6,166,415,648	46.33	0	98.29	94.93
Y-3	20,438,566	6,120,413,870	46.48	0	97.86	93.85

(1) Sample: CK as control line, M as 1-MCP treatment line, Y as 2 mM ethephon treatment line; (2) GC (%): The sum of the number of G and C in the percentage of the total number of bases in high quality reads; (3) Q20 (%): The percentage of bases with a Qphred value of at least 20 in the total number of bases; (4) Q30 (%): The percentage of bases with a Qphred value of at least 30 in the total number of bases.

**Table 2 plants-13-01121-t002:** Gene names of enzymes involved in the starch sucrose metabolic pathway and their replacement names.

Enzyme	Gene ID	Gene ID
SPS	maker-Fvb2-2-augustus-gene-229.38	gene 1
glgC	maker-Fvb6-1-augustus-gene-76.45	gene 2
TPP	maker-Fvb7-3-snap-gene-42.66	gene 3
TPP	maker-Fvb7-1-augustus-gene-276.46	gene 4
beta-amylase	maker-Fvb7-1-augustus-gene-172.32	gene 5
beta-amylase	augustus_masked-Fvb7-3-processed-gene-118.5	gene 6
beta-amylase	maker-Fvb7-2-augustus-gene-115.58	gene 7
beta-amylase	maker-Fvb6-2-augustus-gene-329.38	gene 8
beta-amylase	maker-Fvb5-1-augustus-gene-130.60	gene 9
beta-amylase	maker-Fvb7-4-augustus-gene-113.46	gene 10
endoglucanase	augustus_masked-Fvb7-1-processed-gene-277.6	gene 11
endoglucanase	maker-Fvb2-1-augustus-gene-183.47	gene 12
endoglucanase	maker-Fvb2-3-augustus-gene-13.40	gene 13
MGAM	augustus_masked-Fvb3-3-processed-gene-44.10	gene 14
beta-glucosidase	maker-Fvb3-2-snap-gene-148.16	gene 15
beta-glucosidase	augustus_masked-Fvb3-4-processed-gene-18.7	gene 16
beta-glucosidase	maker-Fvb3-2-augustus-gene-265.38	gene 17
beta-glucosidase	maker-Fvb1-3-snap-gene-210.26	gene 18
beta-glucosidase	maker-Fvb1-3-snap-gene-210.24	gene 19
WAXY	maker-Fvb5-4-augustus-gene-52.55	gene 20
WAXY	snap_masked-Fvb5-3-processed-gene-221.46	gene 21
TPS	augustus_masked-Fvb4-3-processed-gene-14.14	gene 22
TPS	maker-Fvb5-4-augustus-gene-29.85	gene 23
TPS	maker-Fvb4-1-augustus-gene-191.38	gene 24
TPS	augustus_masked-Fvb4-4-processed-gene-21.7	gene 25
pgm	maker-Fvb7-3-augustus-gene-36.50	gene 26
pgm	maker-Fvb7-1-augustus-gene-282.41	gene 27
pgm	maker-Fvb7-3-augustus-gene-8.31	gene 28
EGLC	maker-Fvb1-4-augustus-gene-182.27	gene 29
INV	maker-Fvb7-2-snap-gene-260.73	gene 30
INV	maker-Fvb7-1-augustus-gene-276.47	gene 31
INV	maker-Fvb7-4-snap-gene-32.75	gene 32

**Table 3 plants-13-01121-t003:** Three soluble sugar regression equations and correlation coefficients.

Component	Retention Time/min	Regression Equations	Correlation Coefficient (R^2^)
sucrose	7.6	y = 55,641x + 2215.2	0.992
fructose	9.5	y = 57,315x + 31,904	0.998
glucose	11.5	y = 54,647x + 30,298	0.998

**Table 4 plants-13-01121-t004:** Genetic primers for quantitative real-time PCR.

Gene ID	Forward Primer Sequence (5′–3′)	Reverse Primer Sequence (5′–3′)
gene9	ATGCGCTGCCGAGTATTGCT	ATGCCCTCCATGCGTCACTG
gene10	TAGGAGGTTGGCGGTGGTGA	ATGACACCCAACTGCGGCAA
gene17	ACGTCAAAGATGCGGGTCTGG	ACAGCCACACAGGGAGCAAA
gene27	AGGAACCAGCGGTCTCCGAA	ACCATTCCCTGCGGCGATTT
*FaActin*	TGGGTTTGCTGGAGATGAT	CAGTTAGGAGAACTGGGTGC

## Data Availability

The authors confirm that the data supporting the findings of this study are available within the article.

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
