# Peer review of "Analysis of Ethylene Signal Regulating Sucrose Metabolism in Strawberry Fruits Based on RNA-seq"

_plants, 2024, doi:10.3390/plants13081121_

Round 1

Reviewer 1 Report

Comments and Suggestions for Authors

The manuscript 'Analysis of ethylene signal regulating sucrose metabolism in strawberry fruits based on RNA-seqrevealed by metabolomic and transcriptomic analysis' (plants-2941572) describes the effect of ethylene on the strawberry fruit ripening. The authors used 5 treatments;  3 concentrations of ethephon, 1-MCP and water as a control.

 The manuscript is generally well-prepared, however, key elements of a scientific paper are missing.

 Chapter Material and Methods: Here, the reader should receive precise information on how the experiment was conducted, but this is not the case. There is a lack of information about the cultivar, the procedures for executing individual phases. It is not clear how many repetitions were conducted. Some of this data may be found in other chapters, such as the Results chapter, but all data should be compiled in the Material and Methods chapter. The reader only finds out at the end of the contribution, for example, that 1-MCP is an ethylene inhibitor.

Measurements of pomological properties were conducted 7 days after fruit treatment. Why specifically at that time? Did you do any measurements at maturity?

 Titles of figures and tables do not explain or poorly explain their content.

In the contribution, the authors use many acronyms that are not explained. For example, what do the labels M, Y, GC,... mean in Table 3?

 Description of statistical analyses is missing.

 In many places in the text, there are claims that are not supported by the figures and analysis. For example, in lines 177-178, it is stated that ethephon increased soluble solids, but Figure 1B shows that this is not the case.

 What is soluble sugar content? If it is a sum, it needs to be described.

In Figure 6, the contents of individual sugars are displayed. It would be better to show the composition. The figure should display the sugar composition. It would also be correct to use compositional data analysis.

 Sections 3.2 and 3.3 describe in detail the research work that dealt with genes, but the results are hardly mentioned in the discussion.

Author Response

Q1: The manuscript 'Analysis of ethylene signal regulating sucrose metabolism in strawberry fruits based on RNA-seqrevealed by metabolomic and transcriptomic analysis' (plants-2941572) describes the effect of ethylene on the strawberry fruit ripening. The authors used 5 treatments; 3 concentrations of ethephon, 1-MCP and water as a control.

 The manuscript is generally well-prepared, however, key elements of a scientific paper are missing.

 Chapter Material and Methods: Here, the reader should receive precise information on how the experiment was conducted, but this is not the case. There is a lack of information about the cultivar, the procedures for executing individual phases. It is not clear how many repetitions were conducted. Some of this data may be found in other chapters, such as the Results chapter, but all data should be compiled in the Material and Methods chapter. The reader only finds out at the end of the contribution, for example, that 1-MCP is an ethylene inhibitor.

A1: Thank you for pointing this out. We agree with this comment. Therefore, we have added precise information about the cultivar, the procedures for executing individual phases, the repetitions and so on. It can be found exactly in the Material and Methods (mark revisions in red).

Q2: Measurements of pomological properties were conducted 7 days after fruit treatment. Why specifically at that time? Did you do any measurements at maturity?

A2: In order to investigate the effect of ethylene signal on the ripening and quality formation of strawberry in the fruits' growing stage, strawberry fruits in the white-fruit stage were selected for treatment. By observing the color changes of strawberries, we found that strawberry fruits firstly turned red and entered the mature stage after 7 days of 2mM ETH treatment, and basically turned red and began to enter the mature stage after 7 days of 0.2mM and 10mM ETH treatment, while the fruits of CK and 1-MCP treatment groups were changing color, and strawberry fruits showed significant differences among different treatment groups. With the extension of treatment time, strawberry fruits in the 2mM ETH treatment group showed signs of aging, which was inconsistent with our experimental purpose, so we chose to measure the pomological properties of fruits after 7 days of treatment.

Q3: Titles of figures and tables do not explain or poorly explain their content.

A3: Thank you for pointing this out. We agree with this comment. Therefore, we have modified the titles of figures and tables. It can be found exactly in the Results (mark revisions in red).

Q4: In the contribution, the authors use many acronyms that are not explained. For example, what do the labels M, Y, GC,... mean in Table 3?

A4: Thank you for pointing this out. We agree with this comment. Therefore, we have added a detailed explanation of the acronyms in Table 3 (mark revisions in red). Please see page 6, lines 220-224.

Q5: Description of statistical analyses is missing.

A5: Thank you for pointing this out. We have added the statistical analyses (2.6. statistical analyses) in the Material and Methods (mark revisions in red). Please see page 4, lines 171-174.

Q6: In many places in the text, there are claims that are not supported by the figures and analysis. For example, in lines 177-178, it is stated that ethephon increased soluble solids, but Figure 1B shows that this is not the case.

A6: Thank you for pointing this out. We agree with this comment. Therefore, we have modified them in our new manuscript (mark revisions in red). Please see page 5, lines 185-190, and page 12, lines 328-343.

Q7: What is soluble sugar content? If it is a sum, it needs to be described.

A7: Yes, the soluble sugar content is the sum of sucrose, fructose and glucose. We have added it as you required in our revised MS (mark revisions in red). Please see page 12, lines 329 and 349.

Q8: In Figure 6, the contents of individual sugars are displayed. It would be better to show the composition. The figure should display the sugar composition. It would also be correct to use compositional data analysis.

A8: Agree. We have changed Figure 6 and modified the described as you required in our new MS (mark revisions in red). Please see page 12, lines 328-351.

Q9: Sections 3.2 and 3.3 describe in detail the research work that dealt with genes, but the results are hardly mentioned in the discussion.

A9: Thank you for pointing this out. We agree with this comment. Therefore, we have added them in the Discussion (mark revisions in red). Please see page 13, lines 401-414.

Reviewer 2 Report

Comments and Suggestions for Authors

Dear author's,

Here are the suggestions for improvement: a) Expand the comparative analysis of the results with previous studies on the role of ethylene in fruit ripening. Highlight similarities and differences between the findings of this study and previous research, especially regarding different types of fruits (climacteric versus non-climacteric); b) Discuss the limitations of the present study, such as potential experimental biases, variations in cultivation or analysis conditions, and any knowledge gaps that may have influenced the results; c) Suggest specific areas for future research based on the results of this study. Identify unanswered questions, new hypotheses generated by the findings, or aspects of the phenomenon that warrant further investigation; d) Avoid redundancies and ensure that each point contributes significantly to the overall understanding of the topic. The discussion will become more comprehensive, informative, and convincing, contributing to a deeper understanding of the role of ethylene in strawberry ripening and in non-climacteric fruits in general.

Author Response

Q1: Here are the suggestions for improvement: a) Expand the comparative analysis of the results with previous studies on the role of ethylene in fruit ripening.  Highlight similarities and differences between the findings of this study and previous research, especially regarding different types of fruits (climacteric versus non-climacteric); b) Discuss the limitations of the present study, such as potential experimental biases, variations in cultivation or analysis conditions, and any knowledge gaps that may have influenced the results; c) Suggest specific areas for future research based on the results of this study. Identify unanswered questions, new hypotheses generated by the findings, or aspects of the phenomenon that warrant further investigation; d) Avoid redundancies and ensure that each point contributes significantly to the overall understanding of the topic. The discussion will become more comprehensive, informative, and convincing, contributing to a deeper understanding of the role of ethylene in strawberry ripening and in non-climacteric fruits in general.

A1: Thank you for pointing this out. We agree with this comment. Therefore, we have added or modified them as you required in the Discussion (mark revisions in red). Please see a) page 13, lines 368-374; b) page 14, lines 409-412; c) page 13, lines 391-398; d) page 13 and 14, lines 399-409.

Round 2

Reviewer 1 Report

Comments and Suggestions for Authors The authors have improved the text in an exemplary manner and successfully answered all questions.

Reviewer 2 Report

Comments and Suggestions for Authors

Dear Authors 

I think your article has undergone a real improvement in quality. Congratulations.